# NON-LINEAR NULL SPACE PRIORS FOR INVERSE PROBLEMS

## ABSTRACT

Inverse problems underpin many computational imaging systems, yet are ill-posed due to the nontrivial null-space of the forward operator—signal components that are invisible to the acquisition. Existing methods typically impose generic image priors to regularize these blind directions, but they do not model the null-space structure itself. We introduce a Non-Linear Null-space Prior (NLNP) that jointly learns (i) the image manifold via noise-conditional denoising and (ii) a low-dimensional representation of selected null-space components. Concretely, a network predicts a null-space code from a noisy image, while a measurement encoder predicts the same code from the measurements; at reconstruction time, we penalize the mismatch of the prior and predictor network. Theoretically, we show that training the prior yields a projected Tweedie identity, so the network estimates the projected score of the data distribution, and the resulting regularizer injects orthogonal, state-dependent curvature in the null-space of the sensing matrix, improving conditioning without conflicting with data consistency. We integrate the prior into plug-and-play and validate the approach on compressed sensing and image restoration tasks.

## 1 INTRODUCTION

Inverse problems involve reconstructing an unknown signal from noisy, corrupted, or typically undersampled observations, making the recovery process generally non-invertible and ill-posed. This work focuses on linear inverse problems, where a sensing matrix represents the forward model Bertero et al. (1985). Numerous imaging tasks rely on these principles, including image restoration—such as deblurring, denoising, inpainting, and super-resolution Gunturk & Li (2018)—as well as compressed sensing (CS) Zha et al. (2023); Candes & Wakin (2008) and medical imaging applications like magnetic resonance imaging (MRI) Lustig et al. (2008) or computed tomography (CT) Willemink et al. (2013). See Ongie et al. (2020); Bai et al. (2020); Bertero et al. (2021) and references therein for more applications of imaging inverse problems. Consider the standard formulation of an inverse problem $\boldsymbol{y} = \boldsymbol{H}\boldsymbol{x}^* + \epsilon$, where $\boldsymbol{y} \in \mathbb{R}^m$ denotes the vector of measurements, $\boldsymbol{x}^* \in \mathbb{R}^n$ is the vectorized unknown target signal, $\epsilon$ represents measurement noise and $\boldsymbol{H} \in \mathbb{R}^{m \times n}$ is the linear sensing matrix associated with the acquisition physics, typically with $m \leq n$. Its range-space, denoted Range($\boldsymbol{H}$), comprises all possible measurement vectors $\boldsymbol{y}$ that can be generated by $\boldsymbol{H}\boldsymbol{x}^*$, representing the observable components of the signal captured by the acquisition process. However, this recovery is inherently ill-posed due to the linear operator nature, such as low-dimensionality, which may lead to infinite solutions for $\boldsymbol{x}^*$ that satisfy the observed measurements. Therefore, there is a need to promote prior knowledge about $\boldsymbol{x}^*$, which indicates the structural or statistical properties of the desired signal, enabling recovery for a wider set of measurements by exploiting task-specific characteristics and enhancing solution performance, convergence, and stability in the optimization problem. Hence, selecting a prior tailored to the problem is crucial, as it directly shapes the optimization landscape to yield signal recovery even for components not directly captured by the measurements. To address this, the recovery task is typically formulated as an optimization problem by balancing data fidelity with prior knowledge about the target signal, as follows

$$\hat{\boldsymbol{x}} = \arg\min_{\boldsymbol{x} \in \mathbb{R}^n} g(\boldsymbol{x}) + \lambda h(\boldsymbol{x}), \tag{1}$$

where $g(\cdot)$ is the data fidelity term, generally defined as $||\boldsymbol{H}\boldsymbol{x} - \boldsymbol{y}||_2^2$, $h(\cdot) : \mathbb{R}^n \to \mathbb{R}$ is a regularization function that promotes solutions into an image manifold $\mathcal{M}$, with $\lambda > 0$ controlling the trade-off between both terms. Common priors for inverse problems may include sparsity, modeled by the $\ell_1$-norm Candes & Wakin (2008), promoting solutions with few non-zero components. Other typical priors include total variation (TV) Yuan (2016) to encourage piecewise smoothness, favoring solutions that ensure edge preservation in applications like image deblurring. Further, learning-based methods aim to implicitly learn signal priors from large datasets, leading to more flexible and data-driven solutions. For instance, plug-and-play (PnP) framework Venkatakrishnan et al. (2013); Kamilov et al. (2023) allows the flexible integration of model-based recovery methods with precise forward modeling of the physical acquisition phenomenon with a wide range of data priors. PnP traces back its roots to proximal algorithms Parikh & Boyd (2014), where these operators, usually defined by analytical models of the underlying signals such as sparsity or low-rank Zha et al. (2023), are replaced by a general-purpose image denoiser operator Teodoro et al. (2019).

However, these learned priors typically promote reconstructions that lie within the subspace spanned by clean training data, without explicitly accounting for the null space of the sensing matrix $\boldsymbol{H}$. By constraining solutions to incorporate the null-space of $\boldsymbol{H}$ through a neural network, these priors enable image regularization by explicitly injecting information about the components of $\boldsymbol{x}^*$ that are inherently unobservable in the acquisition process. Thus, these networks exploit the decomposition of a signal into measurement and null-space components, learning a corrective mapping over all null-space modes to enhance interpretability and accuracy Schwab et al. (2019). To improve robustness to measurement noise, Chen & Davies (2020) introduced separate range-space and null-space networks that denoise both components before recombination. Variants of this range-null space decomposition have been applied in diffusion-based restoration Wang et al. (2022); Cheng et al. (2023); Wang et al. (2023b), GAN-prior methods Wang et al. (2023a), algorithm-unrolling architectures Chen et al. (2023), and self-supervised schemes Chen et al. (2025; 2021), consistently leveraging the full null-space projector to achieve high-fidelity reconstructions.

Hence, modeling the null-space remains a fundamental challenge: while range-space priors promote consistency with the measurements, they fail to constrain the unobservable directions in Null($\boldsymbol{H}$), motivating different strategies to regularize these components. In particular, Arguello et al. (2025) represents a linear instance of null-space regularization, where a projection matrix $\boldsymbol{S}$ defines a low-dimensional hyperplane inside Null($\boldsymbol{H}$). Yet, hyperplane estimation is challenging and highly restrictive, as it enforces an overly restrictive linear constraint that may not align with the non-linear data manifold of $\boldsymbol{x}$.

We propose a Non-Linear Null-space Prior (NLNP), a generalization of null-space-based regularization by acknowledging that the data distribution of $\boldsymbol{x}$ typically lies on a non-linear manifold; thus, the prior is not restricted to a hyperplane but instead adapts to the solution space in which $\boldsymbol{x}$ resides. To capture this topology, we introduce a neural network R that learns non-linear projections of $\boldsymbol{S}\boldsymbol{x}$, which (i) relaxes the assumption that Null($\boldsymbol{H}$) can be fully captured by a linear hyperplane $\boldsymbol{S}$ (a constraint that is often too strong and difficult to estimate in practice) and instead replaces it with a flexible non-linear representation, and (ii) aligns this relaxed representation with the image manifold $\mathcal{M}$, by using training with denoising objective. To use this prior on the image reconstruction step, first, a neural network is trained to predict R($\boldsymbol{x}$) using only $\boldsymbol{y}$. We include the prior and the predictor network in a $\ell_2$ regularization. We theoretically analyze that the proposed regularization provides better conditioning in the data-fidelity update and orthogonal curvature towards the image manifold. The proposed NLNP is easily incorporated with common image priors Kamilov et al. (2023). The proposed approach was thoroughly evaluated in two linear inverse problems, Compressed Sensing and image deblurring, showing significant improvements in both convergence and performance.

## 2 NON-LINEAR NULL-SPACE PRIOR

Our key insight is to model, learn, and *use* the geometry of the null-space of the sensing matrix, and to couple it with an image prior learned via denoising. This yields an orthogonal guidance channel in reconstruction: data consistency acts in Range($\boldsymbol{H}^\top$), while the proposed term acts (approximately) in Null($\boldsymbol{H}$).

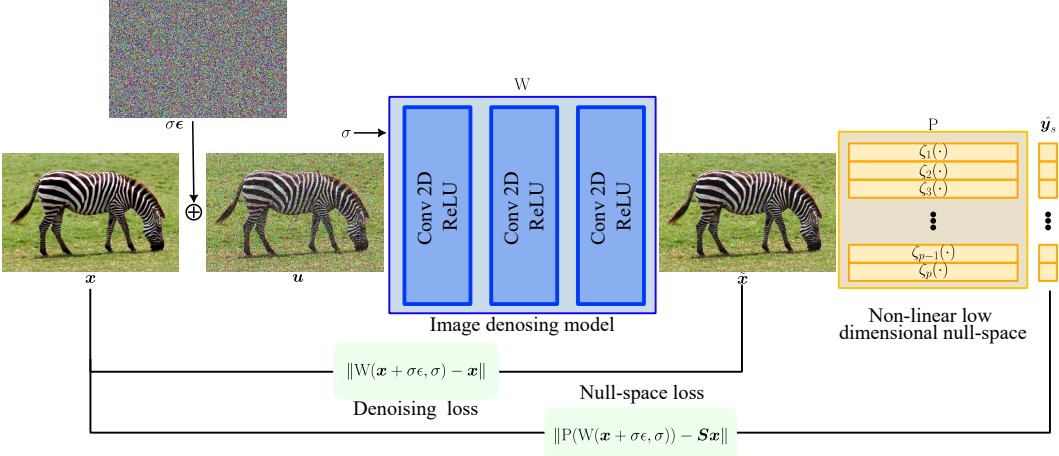

Figure 1: Training scheme for the non-linear null-space prior: noise-conditional image denoising (left head) and null-space code prediction (right head).

**Definition 1 (Null space and projection)** *The null-space of $\boldsymbol{H} \in \mathbb{R}^{m \times n}$ is*

$$\text{Null}(\boldsymbol{H}) = \{\boldsymbol{x} \in \mathbb{R}^n : \boldsymbol{H}\boldsymbol{x} = 0\} = \{\boldsymbol{x} : \boldsymbol{x} \perp \boldsymbol{h}_j, \ \forall j \in \{1, \ldots, m\}\}.$$

*Let $\boldsymbol{P}_n := \boldsymbol{I} - \boldsymbol{H}^\dagger \boldsymbol{H}$ be the orthogonal projector onto $\text{Null}(\boldsymbol{H})$, where $(\cdot)^\dagger$ denotes the Moore–Penrose pseudoinverse.*

Note that in general, every inverse problem solver seeks to determine the null-space component of $\boldsymbol{x}$ only from $\boldsymbol{y}$. However, this is a challenging task. Thus, our prior promotes a solution, not in the *whole* null-space, but rather in low-dimensional projections of it. We select a $p$-dimensional slice of $\text{Null}(\boldsymbol{H})$ via

$$\boldsymbol{S} := \boldsymbol{T}\,\boldsymbol{P}_n \in \mathbb{R}^{p \times n}, \qquad \text{row}(\boldsymbol{S}) \subseteq \text{Null}(\boldsymbol{H}), \tag{2}$$

where the rows of $\boldsymbol{T}$ are orthonormal and constructed from an orthogonal complement of $\text{row}(\boldsymbol{H})$ (QR; see Appx. 6). Then $\boldsymbol{S}\boldsymbol{x}$ provides a compact coordinate of the null-space component of $\boldsymbol{x}$.

### 2.1 Non-Linear Null-Space Prior via Denoising

We propose the model $\text{R}(\cdot, \sigma) : \mathbb{R}^n \times \mathbb{R}_{>0} \to \mathbb{R}^p$ performs a joint image denoising and low-dimensional null-space learning, see Fig. 1 for an illustration. The model needs to be lightweight as it will be used in the image restoration framework (PnP, DM). The model receives as input a noisy image $\boldsymbol{u} = \boldsymbol{x} + \sigma\boldsymbol{\epsilon}$, $\boldsymbol{\epsilon} \sim \mathcal{N}(0, \boldsymbol{I})$ and the noise variance $\sigma$. The first block, $\text{W} : \mathbb{R}^n \times \mathbb{R}_{>0} \to \mathbb{R}^n$, which adapts a standard adaptive noise-level encoding by concatenating the noisy image with $\sigma\mathbf{1}_n$. This first part learns image manifold geometry Milanfar & Delbracio (2025). From the denoised image, a model $\text{P} : \mathbb{R}^n \to \mathbb{R}^p$ performs the low-dimensional null-space estimation. We construct $\text{P} = [\zeta_1(\tilde{\boldsymbol{x}}), \zeta_2(\tilde{\boldsymbol{x}}), \ldots, \zeta_p(\tilde{\boldsymbol{x}})]$, where the non-linear operators are $\zeta_i(\boldsymbol{x}) = \boldsymbol{W}_i^2 \rho(\boldsymbol{W}_i^1 \boldsymbol{x} + \boldsymbol{b}_i^1) + \boldsymbol{b}_i^2, i = 1, \ldots, p$:

$$\text{R}^* = \arg\min_{\text{R}} \ \mathbb{E}_{\boldsymbol{x}, \sigma, \boldsymbol{\epsilon}} \Big[ \underbrace{\|\text{W}(\boldsymbol{x} + \sigma\boldsymbol{\epsilon}, \sigma) - \boldsymbol{x}\|_2^2}_{\text{image denoising}} + \underbrace{\|\text{R}(\boldsymbol{x} + \sigma\boldsymbol{\epsilon}, \sigma) - \boldsymbol{S}\boldsymbol{x}\|_2^2}_{\text{low-dim null-space code}} \Big], \tag{3}$$

with $\boldsymbol{\epsilon} \sim \mathcal{N}(\mathbf{0}, \boldsymbol{I})$ and $\sigma > 0$ drawn from a noise schedule (e.g., log-uniform). The first term forms a standard denoiser W; the second teaches R to predict the null-space components of the clean image from a noisy input. The two-term loss in equation 3 makes $\text{R}(\cdot, \sigma)$ a noise-conditional, manifold-aware null-space encoder. The denoising part learns the local geometry of natural images, while the null-space anchor ties that geometry to a low-dimensional slice $\text{row}(\boldsymbol{S}) \subset \text{Null}(\boldsymbol{H})$. One of the main consequences of this prior is that it learns a *Projected score* of the data. Let $\boldsymbol{u} = \boldsymbol{x} + \sigma\boldsymbol{\epsilon}$ with $\boldsymbol{\epsilon} \sim \mathcal{N}(0, \boldsymbol{I})$. The Bayes-optimal minimizer of the null-space term is the conditional mean

$$\text{R}^\star(\boldsymbol{u}, \sigma) = \mathbb{E}[\boldsymbol{S}\boldsymbol{x} \mid \boldsymbol{u}] = \boldsymbol{S}\boldsymbol{u} + \sigma^2 \boldsymbol{S}\,\nabla_{\boldsymbol{u}} \log p_\sigma(\boldsymbol{u}), \tag{4}$$

i.e., the residual $(\text{R}^\star - \boldsymbol{S}\boldsymbol{u})/\sigma^2$ equals the data score projected onto $\text{row}(\boldsymbol{S})$. This is the exact Tweedie/MMSE identity specialized to $\boldsymbol{S}\boldsymbol{x}$. Another important aspect of this model, as it will

be iteratively used in the reconstruction algorithm, the model requires stable behavior. Following Lipschitz constrained deep denoising Ryu et al. (2019), we apply spectral normalization during training equation 3, i.e.,

$$\boldsymbol{W}_i^1 \leftarrow \frac{\boldsymbol{W}_i^1}{\lambda_{max}(\boldsymbol{W}_i^1)}, \quad \boldsymbol{W}_i^2 \leftarrow \frac{\boldsymbol{W}_i^2}{\lambda_{max}(\boldsymbol{W}_i^2)}, \quad i = 1, \dots p \tag{5}$$

where $\lambda_{max}(\boldsymbol{W})$ denotes the largest eigenvalue of $\boldsymbol{W}$.

## 2.2 MEASUREMENT-CONDITIONED NULL-SPACE PREDICTOR

After pretraining the non-linear prior (Section 2.1), R remains fixed during the subsequent training of a measurement-conditioned encoder $G : \mathbb{R}^m \to \mathbb{R}^p$, designed to predict the same null-space code directly from the measurements $\boldsymbol{y}$. The predictor G bridges the measurement domain and the non-linear null-space representation learned by R, enabling consistency across both perspectives of the inverse problem. Formally, G is trained to minimize the discrepancy between its prediction and the null-space code extracted from the clean ground truth data via R:

$$G^* = \arg\min_{G} \mathbb{E}_{\boldsymbol{x},\boldsymbol{y}} \left\| G(\boldsymbol{y}) - R^*(\boldsymbol{x}, \sigma) \right\|_2^2. \tag{6}$$

This formulation, under Mean Squared Error (MSE), yields $G^*(\boldsymbol{y}) = \mathbb{E}[R(\boldsymbol{x}, \sigma) \mid \boldsymbol{y}] \approx \mathbb{E}[\boldsymbol{S}\boldsymbol{x} \mid \boldsymbol{y}]$, enabling the null-space regularization to adapt to the observed data distribution while maintaining consistency with the image manifold $\mathcal{M}$ induced by R. Since no corrupted image $\boldsymbol{u}$ is leveraged when training G, the $\sigma$ map, which serves as an indicator of the perturbation in the input of R, remains with a zero value. For the architecture of G, any differentiable image-to-image network $E_\theta : \mathbb{R}^n \to \mathbb{R}^n$ that takes the backprojection $x_0 = \boldsymbol{H}^\top \boldsymbol{y}$ and outputs an image-shaped tensor can be used, since the null-space projection is handled by $\boldsymbol{S}$, i.e., $G(\boldsymbol{y}) = \boldsymbol{S} E_\theta(\boldsymbol{H}^\top \boldsymbol{y})$. Thus, G does not need to be tailored to the task.

## 3 REGULARIZED SOLVER

We incorporate the proposed prior using the following optimization problem

$$\hat{\boldsymbol{x}} = \arg\min_{\boldsymbol{x} \in \mathbb{R}^n} \frac{1}{2} \|\boldsymbol{H}\boldsymbol{x} - \boldsymbol{y}\|_2^2 + \lambda\, h(\boldsymbol{x}) + \frac{\gamma}{2} \left\| G^*(\boldsymbol{y}) - R^*(\boldsymbol{x}) \right\|_2^2, \tag{7}$$

with a general image prior $h$ (e.g., sparsity, image denoiser, or diffusion prior). Our framework introduces a novel regularization strategy that embeds data-driven models into inverse-problem solvers by constraining solutions to the nonlinear low-dimensional manifold induced by $R^*$. Note that solving for $\boldsymbol{x}$, the proposed regularization function, requires deriving over the model R. However, due to the spectral normalization induced equation 5 provides stable gradients.

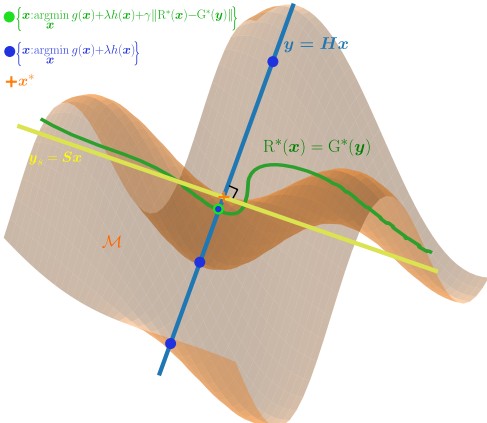

Figure 2: Geometry of the proposed solution. Data-fidelity solution (blue line) The non-linear null-space $\{\boldsymbol{x} : R(\boldsymbol{x}) = G\boldsymbol{y}\}$ curves along the image manifold $\mathcal{M}$ while remaining (approximately) tangent to $\mathrm{Null}(\boldsymbol{H})$, due to range-invariance.

Figure 2 illustrates the geometric interpretation of the proposed non-linear regularized solution. The blue line represents solutions in the $\mathrm{Range}(\boldsymbol{H}^\top)$, the image prior $h(\boldsymbol{x})$ solution represented by the (learned) image manifold $\mathcal{M}$ promoted by PnP denoiser or DM. In the yellow line, the selected null-space component $\boldsymbol{S}\boldsymbol{x}$ is orthogonal to the solution on the $\mathrm{Range}(\boldsymbol{H})$. The proposed prior (green line) approximately follows the direction of the null-space component $\boldsymbol{S}\boldsymbol{x}$ but curves according to the image manifold due to the denoising training scheme. The prior can also be viewed as a relaxation of the null-space hyperplane constraint towards a data-adaptive orthogonal regularization. The proposed solution promotes reconstructions that (i) lie on the image manifold $\mathcal{M}$ (via denoising) and (ii) agree with a measurement-conditioned non-linear null-space code. This resolves the $\mathrm{Null}(\boldsymbol{H})$ ambiguity without fighting data consistency.

## 3.1 PLUG-AND-PLAY METHODS

**Algorithm 1** NLNP-PnP-FISTA

**Require:** $L, \boldsymbol{H}, \boldsymbol{y}, \alpha, \gamma,$
1: $\boldsymbol{x}^0 = \boldsymbol{z}^1 = \frac{1}{2}(\boldsymbol{H}^\top \boldsymbol{y} + \mathrm{JT}(\mathrm{G}^*(\boldsymbol{y}))), \ t = 1$
2: **for** $k = 1, \ldots, L$ **do**
3:     $\sigma_k = \mathrm{scheduler}_\epsilon(k)$
4:     $\boldsymbol{x}^k \leftarrow \boldsymbol{z}^k - \alpha\Big(\boldsymbol{H}^\top(\boldsymbol{H}\boldsymbol{z}^k - \boldsymbol{y}) +$
5:         $\gamma \nabla_{\boldsymbol{z}^k} \|\mathrm{R}^*(\boldsymbol{z}^k, \sigma_k) - \mathrm{G}^*(\boldsymbol{y})\|\Big)$
6:     $\mathbf{x}^k \leftarrow \mathrm{D}_\sigma(\mathbf{x}^k)$
7:     $t' = t$
8:     $t = \frac{1 + \sqrt{1 + 4(t')^2}}{2}$
9:     $\mathbf{z}^{k+1} \leftarrow \mathbf{x}^k + \frac{t'-1}{t}(\mathbf{x}^k - \mathbf{x}^{k-1})$
10: **end for**
11: **return** $\mathbf{x}^k$

The PnP methods Venkatakrishnan et al. (2013); Chan et al. (2016); Kamilov et al. (2023) have gained significant attention due to the flexibility for incorporating physics models and learned image priors. Thus, the term $h(\boldsymbol{x})$ is implicitly solved using a trained denoiser model. We focus on the FISTA-PnP formulation, but it can be easily adapted to other formulations such as HQS or ADMM. In FISTA-PnP, the first step is gradient descent on the data fidelity $g(\boldsymbol{x})$, then apply a denoising step, and finally, an Nesterov acceleration step is used Beck & Teboulle (2009a). In Algorithm 1, the PnP-FISTA with the proposed NLNP regularization is shown. The prior is adapted for each iteration via an estimated noise level with pre-defined $\mathrm{scheduler}_\epsilon(k)$ where $\epsilon$ controls the decay of $\sigma_k$

Note that the proposed regularizer is a non-linear and non-convex function; thus, an accurate algorithm initialization is required. Thus, we set $\boldsymbol{x}^0 \in \mathbb{R}^n$ using the adjoint operator of R. Denote fix a reference point $\tilde{\boldsymbol{u}} \in \mathbb{R}^n$. Define $\tilde{\boldsymbol{v}} := f(\tilde{\boldsymbol{u}}) = R(\tilde{\boldsymbol{u}}, \sigma) \in \mathbb{R}^p$ and the corresponding Jacobian $\mathrm{J}_R(\boldsymbol{x}_0, \sigma) \in \mathbb{R}^{p \times n}$.. Thus, a vector–Jacobian product (VJP) at $\tilde{\boldsymbol{u}}_0$ provides the adjoint map $\mathrm{JT} : \mathbb{R}^p \to \mathbb{R}^n$. Thus, we compute $\mathrm{JT}(\tilde{\boldsymbol{v}}) = \mathrm{J}_R(\tilde{\boldsymbol{u}}, \sigma)^\top \tilde{\boldsymbol{v}}$. Then, the initialization is $\boldsymbol{x}^0 = \frac{1}{2}(\boldsymbol{H}^\top \boldsymbol{y} + \mathrm{JT}(\mathrm{G}^*(\boldsymbol{y})))$

## 3.2 THEORETICAL ANALYSES

FISTA-PnP formulation require gradient steps on data-fidelity and NLNP regularization. Thus, it is required to ensure that the proposed regularization improves i) conditioning algorithm updates and ii) orthogonal curvature in the image manifold. First, let's consider the following assumptions.

(A1) (*Projected Tweedie target*). For $\boldsymbol{u} = \boldsymbol{x} + \sigma\boldsymbol{\epsilon}, \ \boldsymbol{\epsilon} \sim \mathcal{N}(0, \boldsymbol{I})$, $\mathrm{R}^\star(\boldsymbol{u}, \sigma) = \boldsymbol{S}\boldsymbol{u} + \sigma^2 \boldsymbol{S}\nabla_{\boldsymbol{u}} \log p_\sigma(\boldsymbol{u})$ and $J_{\mathrm{R}^\star}(\boldsymbol{u}, \sigma) = \boldsymbol{S}(\boldsymbol{I} + \sigma^2 \nabla_{\boldsymbol{u}}^2 \log p_\sigma(\boldsymbol{u}))$.

(A2) (*Score smoothness*). $\|\nabla_{\boldsymbol{u}}^2 \log p_\sigma(\boldsymbol{u})\| \leq L_\sigma$.

(A3) (*Approximation*). $\|J_{\mathrm{R}} - J_{\mathrm{R}^\star}\| \leq \delta_\sigma$ uniformly near $\boldsymbol{x}_k$.

(A4) (*Range-invariance*). $\|J_{\mathrm{R}}(\boldsymbol{x}, \sigma)\boldsymbol{H}^\top\| \leq \eta$ (with $\eta$ small).

(A5) (*Local regularity*). $J_{\mathrm{R}}$ is $L_J$–Lipschitz and $\|r(\boldsymbol{x}, \sigma)\| \leq \rho$ near $\boldsymbol{x}_k$.

**Theorem 1 (Improved conditioning of linearized $x$-updates)** *Let* $\boldsymbol{H} \in \mathbb{R}^{m \times n}$, *and let* $\boldsymbol{S} \in \mathbb{R}^{p \times n}$ *have orthonormal rows with* $\mathrm{row}(\boldsymbol{S}) \subset \mathrm{Null}(\boldsymbol{H})$. *Define* $U := \mathrm{row}(\boldsymbol{S})$ *with projector* $\boldsymbol{P}_U$. *Consider the regularized objective*

$$F(\boldsymbol{x}) = \tfrac{1}{2}\|\boldsymbol{H}\boldsymbol{x} - \boldsymbol{y}\|_2^2 + \tfrac{\gamma}{2}\|\mathrm{R}(\boldsymbol{x}, \sigma) - \mathrm{G}(\boldsymbol{y})\|_2^2,$$

*and its Gauss–Newton/majorize–minimize $x$-step at $\boldsymbol{x}_k$ with SPD matrix*

$$\boldsymbol{Q}_k = \boldsymbol{H}^\top \boldsymbol{H} + + \gamma J_k^\top J_k, \qquad J_k := J_{\mathrm{R}}(\boldsymbol{x}_k, \sigma).$$

*Let $c_\sigma := 1 - \sigma^2 L_\sigma - \delta_\sigma$ and suppose $c_\sigma > 0$. Then, restricted to $W := \mathrm{Range}(\boldsymbol{H}^\top) \oplus U$,*

$$\lambda_{\min}(\boldsymbol{Q}_k|_W) \geq \min\Big\{\lambda_{\min}(\boldsymbol{H}^\top \boldsymbol{H}), \ \gamma(c_\sigma^2 - \rho L_J)\Big\} - \gamma\eta^2,$$

$$\lambda_{\max}(\boldsymbol{Q}_k|_W) \leq \lambda_{\max}(\boldsymbol{H}^\top \boldsymbol{H}) + \gamma\|J_k\|^2,$$

*and the condition number satisfies*

$$\kappa(\boldsymbol{Q}_k|_W) \leq \frac{\lambda_{\max}(\boldsymbol{H}^\top \boldsymbol{H}) + \gamma\|J_k\|^2}{\min\{\lambda_{\min}(\boldsymbol{H}^\top \boldsymbol{H}), \gamma(c_\sigma^2 - \rho L_J)\} - \gamma\eta^2}.$$

**Remark** *In particular, compared to any fixed linear surrogate ($J_k \equiv S$), the nonlinear R improves the denominator through the state-dependent curvature $c_\sigma^2 - \rho L_J$ on U, yielding fewer CG iterations and larger stable step sizes.*

The proof of this theorem can be found in the Appendix

**Theorem 2 (Orthogonal curvature injection towards the image manifold)** *Under the assumptions of Theorem 1, let $\boldsymbol{x}_\star$ be a stationary point with $R(\boldsymbol{x}_\star, \sigma) = G(\boldsymbol{y})$, and define the* regularizer *field*

$$\mathbf{g}_R(\boldsymbol{x}) := \gamma \, J_R(\boldsymbol{x}, \sigma)^\top \big(R(\boldsymbol{x}, \sigma) - G(\boldsymbol{y})\big).$$

*Then, in a neighborhood of $\boldsymbol{x}_\star$, the following hold.*

*(a) Orthogonality to data consistency. The component of $\mathbf{g}_R(\boldsymbol{x})$ along $\mathrm{Range}(\boldsymbol{H}^\top)$ is small:*

$$\big\| \boldsymbol{P}_{\mathrm{Range}(\boldsymbol{H}^\top)} \, \mathbf{g}_R(\boldsymbol{x}) \big\| \; \leq \; \gamma \, \eta \, \| R(\boldsymbol{x}, \sigma) - G(\boldsymbol{y}) \|,$$

*so the regularizer steers x predominantly inside $\mathrm{Null}(\boldsymbol{H})$ and does not compete with the likelihood gradient $\boldsymbol{H}^\top(\boldsymbol{H}\boldsymbol{x} - \boldsymbol{y}) \in \mathrm{Range}(\boldsymbol{H}^\top)$.*

*(b) Curvature floor in the learned null-space directions. Let $T_{\boldsymbol{x}} := \mathrm{row}\big(J_R(\boldsymbol{x}, \sigma)\big) \subseteq \mathrm{Null}(\boldsymbol{H})$ be the* learned *null-space tangent (at $\boldsymbol{x}$). Then the Hessian of the smooth part of F satisfies*

$$\boldsymbol{P}_{T_{\boldsymbol{x}}} \nabla^2 \Big( \tfrac{1}{2} \| \boldsymbol{H}\boldsymbol{x} - \boldsymbol{y} \|^2 + \tfrac{\gamma}{2} \| r(\boldsymbol{x}, \sigma) \|^2 \Big) \boldsymbol{P}_{T_{\boldsymbol{x}}} \; \succeq \; \gamma \big(c_\sigma^2 - \rho L_J \big) \boldsymbol{P}_{T_{\boldsymbol{x}}} \; - \; \gamma \eta^2 \, \boldsymbol{I},$$

*i.e., along the directions selected by $J_R$ the curvature is uniformly positive (up to $\eta^2$), producing contractive moves towards the agreement set $R(\boldsymbol{x}, \sigma) = G(\boldsymbol{y})$ inside $\mathrm{Null}(\boldsymbol{H})$.*

*(c) Alignment with the image manifold (projected Tweedie). Let $\nabla_{\boldsymbol{u}}^2 \log p_\sigma(\boldsymbol{u})$ have eigen-decomposition with tangent eigenvalues $\{\lambda_t\}$ (near 0) along the smoothed image manifold and normal eigenvalues $\{\lambda_n\} \leq -\kappa$ off-manifold. Then*

$$J_{R^\star}(\boldsymbol{u}, \sigma) = \boldsymbol{S}\big(\boldsymbol{I} + \sigma^2 \nabla_{\boldsymbol{u}}^2 \log p_\sigma(\boldsymbol{u})\big) \quad \Rightarrow \quad \begin{cases} \text{tangent:} & \sigma_{\min}\big(J_{R^\star}\big) \gtrsim 1 - \sigma^2 \max|\lambda_t|, \\ \text{normal:} & \sigma_{\min}\big(J_{R^\star}\big) \geq 1 - \sigma^2 \kappa. \end{cases}$$

*Thus $J_{R^\star}^\top J_{R^\star}$ is* stronger *in normal (off-manifold) directions than in tangent ones, so the induced curvature selectively pulls the iterate towards the image manifold while remaining in $\mathrm{Null}(\boldsymbol{H})$. The same conclusions hold for $J_R$ up to the approximation error $\delta_\sigma$.*

The proof of this theorem can be found in the Appendix

## 4 EXPERIMENTS

We evaluate the proposed NLNP regularizer on two imaging inverse problems: Compressed Sensing (CS) and deblurring. For the recovery stage, we employ the FISTA solver Beck & Teboulle (2009b) within a PnP framework equipped with a deep denoiser Kamilov et al. (2023), regularization by denoising (RED) Romano et al. (2017), and a sparsity prior Beck & Teboulle (2009b). The multi-term data-fidelity weighting into the FISTA-PnP algorithm includes an acceleration factor that follows a two-phase, piecewise-constant schedule: denoting by $\gamma_k$ the scaling applied to the NLNP-driven fidelity term at iteration $k$, then

$$\gamma_k = \begin{cases} \gamma_{\text{high}}, & 0 \leq k < k^*, \\ \gamma_{\text{low}}, & k \geq k^*, \end{cases} \tag{8}$$

with $k^* = K/2$, where $K$ is the total number of iterations. This transition, rather than a gradual decay, emphasizes aggressive enforcement of the non-linear consistency constraints during the initial phase before reducing their influence to allow the remaining prior terms to refine fine-scale details with improved stability. The method was implemented using the PyTorch framework. All experiments were conducted on an NVIDIA TITAN RTX GPU.

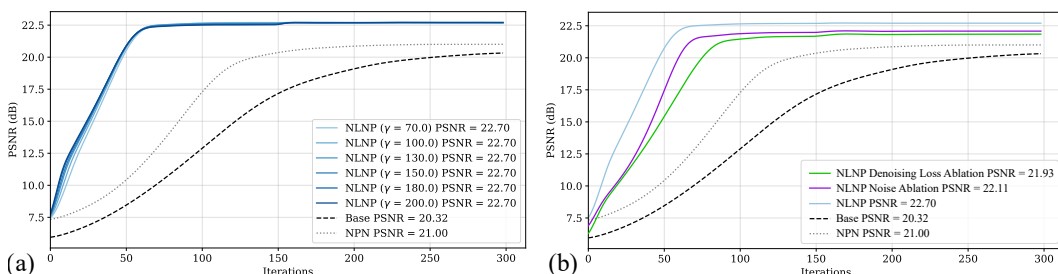

Figure 3: Compressed Sensing (CS) PSNR results. Left: PSNR for sampling ratios $m/n = 0.1$ and $p/n = 0.3$. Right: overlay of the best factor among the left-panel settings against two ablation studies—(i) the noise used during R pretraining and (ii) the denoising loss used in that stage—. The solver and acquisition operator are kept fixed; only the training configuration and architecture of R varies.

**Compressed Sensing.** The Single-Pixel Camera (SPC) was used along with the CelebA dataset Liu et al. (2015), with 16,000 images for training and 4,000 for testing. All images were resized to $32 \times 32$. The Adam optimizer Kingma & Ba (2014) was used with a learning rate of $3 \times 10^{-4}$ and a batch size of 256. $\boldsymbol{H}$ is a random binary sensing matrix with $m/n = 0.1$ and $p/n = 0.3$ while $\boldsymbol{S}$ is initialized by QR decomposition, according to Algorithm 1 in Appendix A.1. Then, G was implemented using a U-Net architecture Ronneberger et al. (2015), and in this setting $\boldsymbol{S}$ may serve as the fixed null-space projection matrix obtained from the QR decomposition. For the implementation of the $\gamma$ scheduler, we used $\gamma_{\text{high}} = \{70, 100, 130, 150, 180, 200\}$ and $\gamma_{\text{low}} = 70$.

## 4.1 CONVERGENCE AND PERFORMANCE ANALYSIS IN SPC

We begin by assessing convergence speed and stability under SPC via iteration-wise PSNR trajectories across methods and hyperparameters. Specifically, Fig. 3(a) reports the test-set PSNR versus NPN (linear case) and FISTA-PnP (Base) iteration for the sampling ratios crA = 0.1 and crB = 0.3, using the same denoiser prior. Across all $\gamma$ cases, the NLNP solver converges in fewer iterations and exhibits reduced oscillations; the inflection near $k^*$ aligns with the scheduled reduction of the NLNP weight, after which the remaining priors refine fine-scale details with improved stability. The $\gamma$ value for the NPN method remains fixed at $\gamma = 1$ since it was determined to be its optimal value.

Furthermore, to numerically measure the convergence improvement obtained via NLNP, we track reconstruction quality as a function of solver iterations under the SPC setting. Complementing the PSNR traces, the convergence plot in Fig. 4 reports a normalized one-step contraction metric (values below 1 indicate contraction and values approaching 1 indicate the asymptotic regime) for NLNP with multiple factors, the Base FISTA solver, and the linear NPN variant. All NLNP configurations exhibit two pronounced contraction windows at the initial phase within the first iterations and a second phase around the schedule transition at $k^*$, followed by a monotone approach to the fixed point with minimal oscillations. In contrast, the Base method displays a shallower and slower contraction, whereas NPN shows a transient overshoot and delayed stabilization near the mid-iteration transition. The depth of the early contraction and the speed of stabilization are consistently stronger under NLNP, indicating faster error reduction per iteration.

## 4.2 ABLATION STUDIES

To further analyze and substantiate the contribution of our pretraining choices to improve the effectiveness of NLNP, we conducted two ablation studies designed to isolate the roles of noise conditioning and the denoising objective in R. The objective is to corroborate that these design elements improve both reconstruction performance and the convergence behavior of the regularizer. (i) In the first ablation, we removed the denoising loss used during the pretraining of R (equation 3) while retaining the addition of Gaussian noise (architecture unchanged). (ii) In the second ablation, we removed both the noise injection and the denoising loss; to eliminate implicit denoising capacity, we also modified R by removing the denoising block $W$, yielding an otherwise comparable architecture trained on the same data. For each variant, we retrained G and R, while the reconstruction

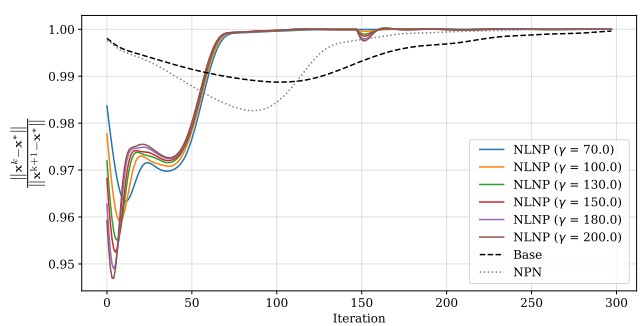

Figure 4: Compressed Sensing (CS) convergence results

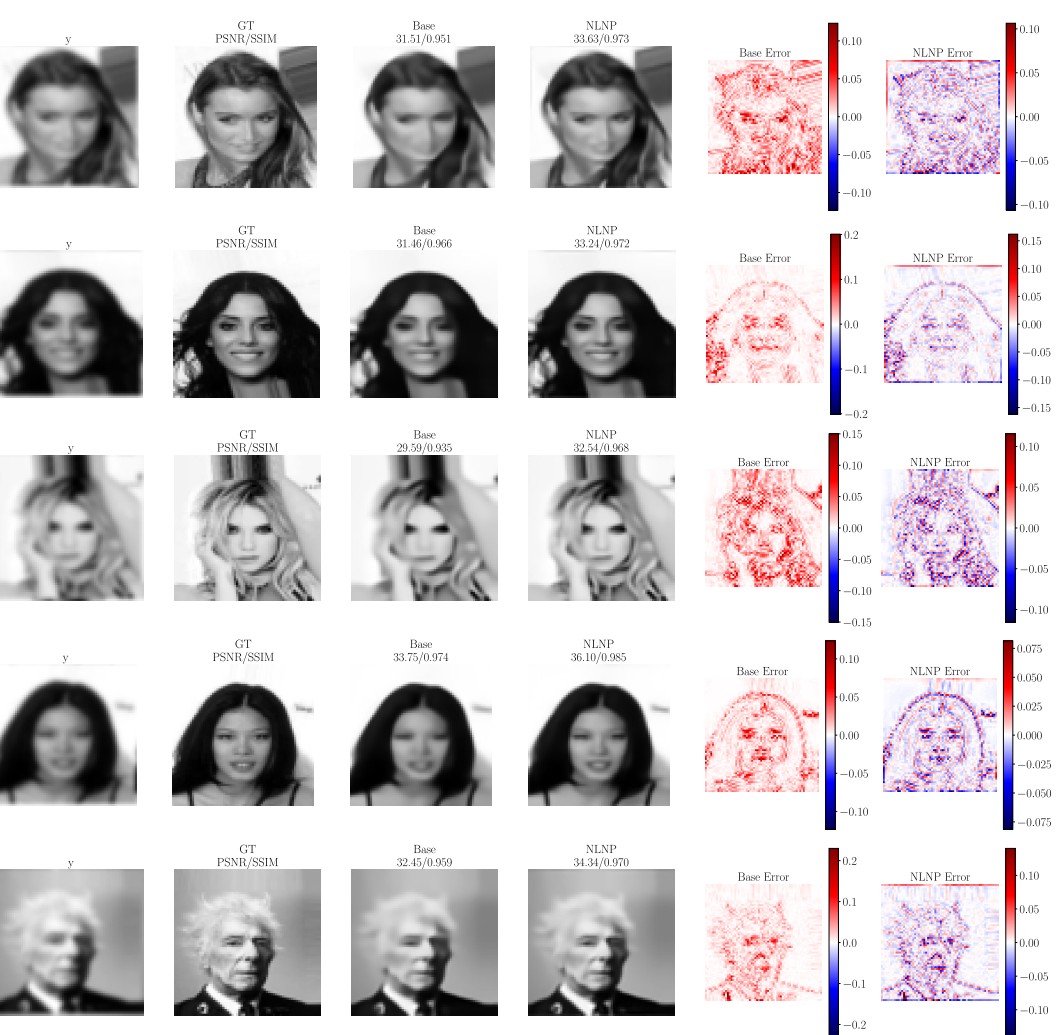

Figure 5: Visual reconstructions for image deblurring for baseline FISTA PnP and NLPN

solver, datasets, and all downstream hyperparameters were kept fixed. Both ablations exhibit slower PSNR ascent and lower terminal PSNR, with the strongest degradation observed when both noise conditioning and denoising are removed.

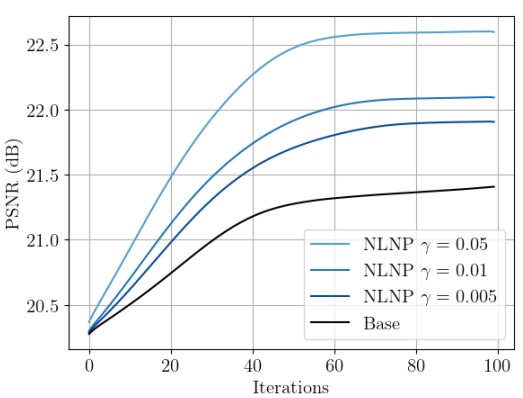

Figure 6: FISTA-PnP Recovery for baseline (no regularization) and NLNP for different values of $\gamma$

**Deblurring.** In the deblurring setting, we applied a 2-D Gaussian kernel with a bandwidth of $5.0\,\sigma$. Experiments were conducted on the CelebA dataset resized to $64 \times 64$, using 16,000 images for training and 2,000 for testing. The measurement-conditioned network G we employed a U-Net architecture along with the Adam optimizer with a learning rate of $1 \times 10^{-4}$ and a batch size of 32. For these experiments, we set $p/n = 0.1$. The $\gamma$ scheduler was implemented by using $\gamma_{\text{high}} = \{0.5, 0.01, 0.005\}$ and $\gamma_{\text{low}} = 0.001$. The results show significant improvements of up to 1 dB in recovery perfomance and improved convergences as predicted with Theorem 1. In Figure 5 is shown some visual results of the proposed method are shown compared with baseline PnP-FISTA for a Gaussian kernel bandwidth of 1.

## 5 LIMITATIONS

While NLNP offers a principled means to inject measurement–conditioned, non-linear null-space structure into inverse problems, its applicability is limited by its reliance on an accurately specified linear forward model. In particular, the method requieres a calibrated sensing operator $\boldsymbol{H}$ and access to a null-space slice $\text{row}(\boldsymbol{S}) \subseteq \text{Null}(\boldsymbol{H})$ (either fixed from a QR/SVD factorization or jointly updated via $\boldsymbol{S} \leftarrow \boldsymbol{S}(\boldsymbol{I} - \boldsymbol{H}^\dagger \boldsymbol{H})$). The range-invariance used in training/analysis and the enforcement of null-space consistency depend on the fidelity of $\boldsymbol{H}$; consequently, model mismatch, spatial variation, or miscalibration can degrade performance and weaken the geometric guarantees. Furthermore, NLNP is highly sensitive to the representation and selection of the projection $\boldsymbol{S}$, which encodes a representational choice for the subspace $U = \text{row}(\boldsymbol{S}) \subseteq \text{Null}(\boldsymbol{H})$. Since R is trained to predict the projected code $\boldsymbol{Sx}$, its usability hinges on the design of $\boldsymbol{S}$. Hence, misalignment of $U$ with task-relevant null-space modes, perturbations induced by range-domain leakage into $\boldsymbol{S}$ (i.e. non-orthonormal rows) or poor conditioning can bias the learned null-space code towards uninformative directions and reduce the benefits of NLNP.

Moreover, since NLNP constrains reconstructions through a selected low-dimensional subspace $\text{row}(\boldsymbol{S})$ of $\text{Null}(\boldsymbol{H})$, the selection of the null-space slice dimension $p$ remains a key point in the learning process. The slice dimension $p$ comprehends a trade-off between the stability and expressivity of $\text{Null}(\boldsymbol{H})$: if $p$ is too small, allowable variability in the null-space may be poorly represented and subsequently learned by R, leading to a highly constrained solution space imposed by the learned regularizer G, which may lead to suboptimal solutions; alternatively, when employing a large $p$, the constraint tends to become weak, decreasing optimization performance and reducing useful residual ambiguity. Hence, developing data- and operator-aware procedures to select or learn $\boldsymbol{S}$ (including automatic choice of $p$) is an important direction for future work.

## 6 CONCLUSIONS AND FUTURE OUTLOOKS

We introduced a *Non-Linear Null-space Prior* (NLNP) that learns a low-dimensional, measurement-conditioned representation of $\text{Null}(H)$ and couples it with a noise-conditional denoising head. The resulting regularizer injects *orthogonal, state-dependent curvature* in directions invisible to the sensor while remaining nearly orthogonal to $\text{Range}(H^\top)$. Theoretically, we proved (i) improved conditioning of Gauss–Newton $x$-updates through a curvature floor on the learned null-space slice, and (ii) orthogonal curvature injection towards the image manifold, which together explain the faster and more stable convergence observed in our experiments. Empirically, integrating NLNP into PnP solvers yields consistent gains in PSNR and iteration efficiency across compressed sensing and deblurring. The prior can also be included in other solvers, such as deep unfolding, diffusion models, and deep equilibrium models.

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

APPENDIX

### A.1 ALGORITHMS FOR DESIGNING $S$

We developed an algorithm for obtaining $S$ from $H$ satisfying that $SH^\top = 0$. The algorithm is based on the QR decomposition, first computing a full QR decomposition of $H^\top \in \mathbb{R}^{n \times m}$, yielding an orthonormal basis $Q_{\text{full}} \in \mathbb{R}^{n \times n}$ for $\mathbb{R}^n$. The columns from $m+1$ to $n$ of $Q_{\text{full}}$ form a basis for $\text{Null}(H)$, which we denote by $N \in \mathbb{R}^{n \times (n-m)}$. To construct a subspace of the null space, the algorithm samples a random Gaussian matrix $P \in \mathbb{R}^{(n-m) \times p}$, which is orthonormalized via QR decomposition to produce $U \in \mathbb{R}^{(n-m) \times p}$. This ensures that the resulting subspace is both diverse and well-conditioned. Finally, the matrix $S$ is obtained as $S = U^\top N^\top \in \mathbb{R}^{p \times n}$, which consists of $p$ orthonormal vectors that span a random $p$-dimensional subspace within $\text{Null}(H)$.

---

**Algorithm 2** Generate orthonormal rows to $H$ via QR decomposition

---

**Require:** Matrix $H \in \mathbb{R}^{m \times n}$, desired number of rows $p$
**Ensure:** Matrix $S \in \mathbb{R}^{p \times n}$ whose rows are orthonormal and lie in $\text{Null}(H)$
1: $Q_{\text{full}} \leftarrow \text{QR}(H^\top)$
2: $N \leftarrow Q_{\text{full},\, m+1:n}$          ▷ Nullspace basis, size $n \times (n-m)$
3: Sample $P \sim \mathcal{N}(0, I) \in \mathbb{R}^{(n-m) \times p}$
4: $U \leftarrow \text{QR}(P)$          ▷ $U \in \mathbb{R}^{(n-m) \times p}$ with orthonormal columns
5: $S \leftarrow U^\top N^\top$          ▷ Resulting $p \times n$ matrix of orthonormal rows
6: **return** $S$

---

### A.2 PROOF OF THEOREM 1

**Proof** Denote the residual $r(\boldsymbol{x}, \sigma) := \text{R}(\boldsymbol{x}, \sigma) - \text{G}(\boldsymbol{y}) \in \mathbb{R}^p$ and $J(\boldsymbol{x}, \sigma) := J_\text{R}(\boldsymbol{x}, \sigma)$, and abbreviate $r_k := r(\boldsymbol{x}_k, \sigma)$, $J_k := J(\boldsymbol{x}_k, \sigma)$. Let $V := \text{Range}(H^\top)$, so $W = V \oplus U$ with $V \perp U$.

Under (A1)–(A3), for any $u \in U$ we have
$$\|J_k u\| \geq \|J_{\text{R}^\star} u\| - \|J_k - J_{\text{R}^\star}\| \|u\| \geq \left(1 - \sigma^2 L_\sigma - \delta_\sigma\right) \|u\| = c_\sigma \|u\|.$$
Thus $u^\top J_k^\top J_k u = \|J_k u\|^2 \geq c_\sigma^2 \|u\|^2$.

The exact Hessian of the residual penalty satisfies

$$\nabla^2\!\left(\tfrac{\gamma}{2} \|r(\boldsymbol{x}, \sigma)\|^2\right) = \gamma\, J(\boldsymbol{x}, \sigma)^\top J(\boldsymbol{x}, \sigma) + \gamma \sum_{i=1}^{p} r_i(\boldsymbol{x}, \sigma)\, \nabla^2 \text{R}_i(\boldsymbol{x}, \sigma).$$

By (A5), $\|r(\boldsymbol{x}, \sigma)\| \leq \rho$ and $J$ is $L_J$–Lipschitz, so $\|\nabla^2 \text{R}_i\| \leq L_J$. Hence, at $\boldsymbol{x}_k$,

$$\gamma\, J_k^\top J_k \succeq \nabla^2\!\left(\tfrac{\gamma}{2} \|r(\boldsymbol{x}_k, \sigma)\|^2\right) - \gamma \rho L_J\, \boldsymbol{I}. \tag{9}$$

Assumption (A4) yields $\|J(\boldsymbol{x}, \sigma) H^\top\| \leq \eta$, hence $\|J_k P_V\| \leq \eta$ with $P_V$ the projector onto $V$. Using the $V \oplus U$ splitting,

$$\| P_V J_k^\top J_k P_U \| = \|(J_k P_V)^\top (J_k P_U)\| \leq \|J_k P_V\| \|J_k P_U\| \leq \eta \|J_k P_U\|.$$

Since $\|J_k P_U\|$ is uniformly bounded near $\boldsymbol{x}_k$ (by (A3) and Step 1), we absorb this harmless constant into $\eta$ and write $\|P_V J_k^\top J_k P_U\| \leq \eta^2$. Therefore, for any unit $z = v + u$ with $v \in V$, $u \in U$,

$$z^\top (\gamma J_k^\top J_k) z \geq \gamma\, u^\top (P_U J_k^\top J_k P_U) u - \gamma \eta^2. \tag{10}$$

Combining equation 9, Step 1, and equation 10 gives

$$z^\top (\gamma J_k^\top J_k) z \geq \gamma\left(c_\sigma^2 - \rho L_J\right) \|u\|^2 - \gamma \eta^2. \tag{11}$$

Then, for any unit $z = v + u \in W$,

$$z^\top \boldsymbol{Q}_k z = v^\top H^\top H\, v + z^\top (\gamma J_k^\top J_k) z$$
$$\geq \lambda_{\min}(H^\top H) \|v\|^2 + \gamma\left(c_\sigma^2 - \rho L_J\right) \|u\|^2 - \gamma \eta^2.$$

As $\|z\|^2 = \|v\|^2 + \|u\|^2 = 1$, we obtain

$$\lambda_{\min}\big(\boldsymbol{Q}_k|_W\big) \geq \min\left\{\lambda_{\min}(\boldsymbol{H}^\top\boldsymbol{H}), +\gamma(c_\sigma^2 - \rho L_J)\right\} - \gamma\eta^2.$$

For any unit $z \in W$,

$$z^\top \boldsymbol{Q}_k z \leq \lambda_{\max}(\boldsymbol{H}^\top\boldsymbol{H}) + \gamma\|J_k\|^2,$$

hence

$$\lambda_{\max}\big(\boldsymbol{Q}_k|_W\big) \leq \lambda_{\max}(\boldsymbol{H}^\top\boldsymbol{H}) + \gamma\|J_k\|^2.$$

Combining the bounds yields

$$\kappa\big(\boldsymbol{Q}_k|_W\big) \leq \frac{\lambda_{\max}(\boldsymbol{H}^\top\boldsymbol{H}) + \gamma\|J_k\|^2}{\min\{\lambda_{\min}(\boldsymbol{H}^\top\boldsymbol{H}), +\gamma(c_\sigma^2 - \rho L_J)\} - \gamma\eta^2}.$$

This completes the proof.

### A.3 PROOF OF THEOREM 2

**Proof.** Set $r(\boldsymbol{x}, \sigma) := \mathrm{R}(\boldsymbol{x}, \sigma) - \mathrm{G}(\boldsymbol{y})$ and $J(\boldsymbol{x}, \sigma) := J_\mathrm{R}(\boldsymbol{x}, \sigma)$; abbreviate $r := r(\boldsymbol{x}, \sigma)$, $J := J(\boldsymbol{x}, \sigma)$. Let $V := \mathrm{Range}(\boldsymbol{H}^\top)$ and $U := \mathrm{row}(\boldsymbol{S}) \subset \mathrm{Null}(\boldsymbol{H})$ with orthogonal projectors $\boldsymbol{P}_V, \boldsymbol{P}_U$. Define $c_\sigma := 1 - \sigma^2 L_\sigma - \delta_\sigma > 0$ as in Theorem 1.

**(a) Orthogonality of the regularizer field.** By definition, $\mathrm{g}_\mathrm{R}(\boldsymbol{x}) = \gamma J^\top r$. Then

$$\|\boldsymbol{P}_V \mathrm{g}_\mathrm{R}(\boldsymbol{x})\| = \gamma \sup_{\substack{v \in V \\ \|v\|=1}} \langle v, J^\top r \rangle = \gamma \sup_{\substack{v \in V \\ \|v\|=1}} \langle Jv, r \rangle \leq \gamma \|J|_V\| \, \|r\|.$$

Assumption (A4) ("range-invariance") gives $\|J(\boldsymbol{x}, \sigma)\boldsymbol{H}^\top\| \leq \eta$, which we interpret as a bound on the restriction of $J$ to $V$ (absorbing harmless constants into $\eta$). Hence $\|\boldsymbol{P}_V \mathrm{g}_\mathrm{R}(\boldsymbol{x})\| \leq \gamma\eta\|r\|$, proving (a).

**(b) Curvature floor along $T_{\boldsymbol{x}} = \mathrm{row}(J)$.** We first obtain a singular-value floor for $J$ on its row space. From (A1)–(A2),

$$J_{\mathrm{R}^\star}(\boldsymbol{x}, \sigma) = \boldsymbol{S}\big(\boldsymbol{I} + \sigma^2 \nabla^2 \log p_\sigma(\boldsymbol{x})\big) =: \boldsymbol{S}K_{\boldsymbol{x}}, \quad \text{so} \quad \sigma_{\min}^+(J_{\mathrm{R}^\star}) \geq 1 - \sigma^2 L_\sigma.$$

By (A3), $\|J - J_{\mathrm{R}^\star}\| \leq \delta_\sigma$; Weyl's inequality then yields

$$\sigma_{\min}^+(J) \geq \sigma_{\min}^+(J_{\mathrm{R}^\star}) - \delta_\sigma \geq c_\sigma.$$

Therefore, for any $z \in T_{\boldsymbol{x}} = \mathrm{Range}(J^\top)$,

$$z^\top J^\top J z = \|Jz\|^2 \geq c_\sigma^2 \|z\|^2. \tag{12}$$

Next, use the Gauss–Newton identity and (A5):

$$\nabla^2\big(\tfrac{\gamma}{2}\|r\|^2\big) = \gamma J^\top J + \gamma \sum_{i=1}^p r_i \nabla^2 R_i \succeq \gamma J^\top J - \gamma \rho L_J \boldsymbol{I}.$$

Projecting onto $T_{\boldsymbol{x}}$ and combining with equation 12 gives

$$\boldsymbol{P}_{T_{\boldsymbol{x}}} \nabla^2\big(\tfrac{\gamma}{2}\|r\|^2\big) \boldsymbol{P}_{T_{\boldsymbol{x}}} \succeq \gamma\big(c_\sigma^2 - \rho L_J\big) \boldsymbol{P}_{T_{\boldsymbol{x}}}.$$

Adding the data Hessian $\nabla^2(\tfrac{1}{2}\|\boldsymbol{H}\boldsymbol{x} - \boldsymbol{y}\|^2) = \boldsymbol{H}^\top\boldsymbol{H} \succeq 0$ only helps. Finally, to account for small range-leakage implied by (A4) when regarding $T_{\boldsymbol{x}}$ as (approximately) contained in $\mathrm{Null}(\boldsymbol{H})$, we insert the slack $-\gamma\eta^2\boldsymbol{I}$, obtaining the stated bound in (b).

**(c) Manifold-aligned anisotropy (projected Tweedie).** Let $\nabla_{\boldsymbol{u}}^2 \log p_\sigma(\boldsymbol{u}) = \boldsymbol{Q}\Lambda\boldsymbol{Q}^\top$ with eigenvalues $\{\lambda_t\}$ on manifold tangents and $\{\lambda_n\} \leq -\kappa$ on normals. From (A1),

$$J_{\mathrm{R}^\star}(\boldsymbol{u}, \sigma) = \boldsymbol{S}\big(\boldsymbol{I} + \sigma^2 \nabla_{\boldsymbol{u}}^2 \log p_\sigma(\boldsymbol{u})\big) = \boldsymbol{S}\boldsymbol{Q}\big(\boldsymbol{I} + \sigma^2\Lambda\big)\boldsymbol{Q}^\top.$$

Along an eigen-direction $v$ with eigenvalue $\lambda$, $\|J_{\mathrm{R}^\star}v\| = \|\boldsymbol{S}(\boldsymbol{I} + \sigma^2\lambda)v\| = |1 + \sigma^2\lambda|\,\|\boldsymbol{P}_U v\|$. Hence, on tangents ($\lambda = \lambda_t \approx 0$), $\sigma_{\min}(J_{\mathrm{R}^\star}) \gtrsim 1 - \sigma^2 \max|\lambda_t|$, while on normals ($\lambda = \lambda_n \leq -\kappa$), $\sigma_{\min}(J_{\mathrm{R}^\star}) \geq 1 - \sigma^2\kappa$. Thus $J_{\mathrm{R}^\star}^\top J_{\mathrm{R}^\star}$ produces stronger contraction off-manifold than along it, guiding iterates towards the image manifold while remaining inside $\mathrm{Null}(\boldsymbol{H})$ due to the projection $\boldsymbol{S}$. By (A3), the same holds for $J$ up to the approximation error $\delta_\sigma$, which completes (c).

## A.4 THE USE OF LARGE LANGUAGE MODELS

Large Language Models (LLMs), such as ChatGPT Pro, were used by the authors to receive grammar, style, and clarity suggestions on author-written text, and all edits were reviewed and either accepted or rejected by the authors. Also, it was used for formalize the proofs of Theorems 1 and 2, which were thoroughly checked by the authors. No LLM-generated text, data, analyses, code, or references were accepted without human verification, and the authors take full responsibility for the paper's content.

