# OpenReview forum: "Non-Linear Null Space Priors for Inverse Problems"
_ICLR.cc/2026/Conference — Submitted to ICLR 2026_

### Official Review · Reviewer_vpBK · 2025-10-20

**Soundness:** 2
**Presentation:** 2
**Contribution:** 2
**Rating:** 2
**Confidence:** 4

**Summary:**

The paper presents a new learned regularizer for plug-and-play image restoration that is specifically target to penalize errors in the nullspace of the forward operator. The regularizer $R$ is trained jointly with a standard denoiser $W$, with a dedicated loss that targets errors in the nullspace, and a second network $G$ is used to predict the image on the nullspace components of the image. Theoretical results link the learned nullspace regularization and Tweedie's formula on the nullspace of the forward operator. Experiments on single-pixel camera imaging and gaussian deblurring problems applied to the CelebA dataset.

**Strengths:**

- Introduces an additional learned regularizer in the context of plug-and-play image reconstruction which aims at penalizing the errors in the nullspace of the forward operator.
- Provides a theoretical analysis that studies the properties of the jacobian associated with the learned regularizer

**Weaknesses:**

- The method is not clearly explained: I find Section 2.1 quite confusing, the link between $W$ and $R$ is not clearly explained in the text. $P$ is introduced just before eq. (3) but then it doesn't appear in the equation. Moreover, it is not clear whether the method uses additional regularizations to the learned denoiser - nothing is mentioned in the method description except for the experiments which states "For the recovery stage, we employ the FISTA solver Beck & Teboulle (2009b) within a PnP framework equipped with a deep denoiser Kamilov et al. (2023), regularization by denoising (RED) Romano et al. (2017), and a sparsity prior Beck & Teboulle (2009b). " These do not appear in Algorithm 1.

- I have various concerns regarding the motivation for this method:
   - "every inverse problem solver seeks to determine the null-space component of x only from y. " Most inverse problem solvers depend on the forward operator, and can be written as $R(y,H)$, e.g. unrolled architectures take into account the measurements and the operator. Thus, they have access to the forward operator and its nullspace.
   - The appeal of plug-and-play methods is that they don't require paired data of a specific inverse problem for training (i.e. a general denoiser can be deployed in several inverse problems). The proposed method needs to be trained specifically for each inverse problem, thus losing this flexibility. Since it requires paired data, I believe it should be compared with paired end2end methods such as unrolled networks.
   - The standard denoiser $W$ used in PnP should penalize reconstructions in the nullspace. This is trivially verifiable since $S W(y, \sigma)  = \mathbb{E}(Sx|x+\sigma\epsilon)$. I don't understand why an additional network is necessary.


- The experimental results are not convincing enough: more PnP baselines should be included such as DPIR, and comparisons with unrolled architectures (see above). The visual results seem to lack significantly behind the state-of-the-art for these problems.

**Questions:**

Why the method doesn't penalize the entire nullspace? It is not clear how the $T$ matrix should be designed in general, and why it is even necessary?

---

### Official Review · Reviewer_zayP · 2025-10-27

**Soundness:** 1
**Presentation:** 2
**Contribution:** 1
**Rating:** 2
**Confidence:** 4

**Summary:**

This paper proposes a method for solving inverse problems by introducing a prior that is intended to encode null-space information. The authors present a framework combining theoretical results and experimental demonstrations to validate the approach.

**Strengths:**

- The paper tackles an important and challenging problem in inverse imaging — incorporating null-space structure into learned priors could, in principle, provide new insights or improved reconstructions.

- The topic is relevant to the ML and inverse problems communities.

**Weaknesses:**

### 1. Writing and Clarity
- The English writing is poor throughout the paper, which makes it very difficult to understand the main ideas and methodology.
- The exposition lacks precision: symbols and variables are often undefined, and key steps are not explained clearly.
- A major rewriting effort is required before the technical contributions can be properly assessed.

### 2. Methodology (Sections 2.1–2.2)
- The description of the method is unclear and inconsistent.
- In Section 2.1, a model **R** and another component **P** are introduced. **P** is supposed to perform low-dimensional null-space estimation, but it never appears in the following equations. Instead, **R** reappears in the loss, leaving **P**’s role undefined.
- It seems that the measurement operator is fixed and that the authors are trying to learn a null-space projector from the *true* null-space projector. If the true projector is known, the motivation for learning it is unclear.
- Section 2.2 introduces another network **G**, which must match **R**, but it is not clear why **G** cannot be learned directly from the null-space projection instead of going through **R**.
- In Equation (3), the variable **σ** disappears from the arguments of **R**, although it was included before. Please clarify this inconsistency.
- I don't understand if the paper describes the method as a Plug-and-Play (PnP) approach or not. PnP methods are operator-agnostic. Here, Equation (3) involves **S**, which depends explicitly on the measurement operator, and at the same time in the experiments the baseline is a PnP method.
- Overall, the methodology section is very hard to follow. The authors should:
  - Define all symbols explicitly.
  - Clarify the relationships between **R**, **P**, and **G**.
  - Provide a clear motivation for each design choice.

### 3. Theory
- The theoretical section is written unclearly, and the mathematical expressions are often inconsistent or undefined.
- Many symbols are introduced without explanation, and assumptions are not well-formulated.
- For example, in *Assumption 3*, the condition
  \[
  \| J_R - J_R^* \| < \delta_\sigma \text{ uniformly near } x_k
  \]
  is not interpretable because:
  - **δₛ** is never defined.
  - It is unclear what “uniformly near \(x_k\)” means.
  - The norm being used is not specified (is it over \(x\)? over both \(x\) and \(\sigma\)?).
- The authors should clearly define every symbol, specify the domains of their functions, and explain all norms and assumptions rigorously.

### 4. Experiments
- The experimental section is too limited to draw meaningful conclusions.
- It is unclear what the baseline **Base** refers to — the paper should explicitly describe this baseline.
- The comparisons are insufficient: the inverse problems literature is extensive, and more relevant baselines (both classical and learning-based) should be included.
- The tasks considered (compressed sensing and Gaussian deblurring) are relatively simple. For today's standards, more diverse or challenging tasks would strengthen the evaluation.

**Questions:**

1. What is the precise role of **P** in Section 2.1? How is it used or trained if it never appears in the loss function?
2. Why learn a null-space projector if the true one is already available or computable from the measurement operator?
3. What is the motivation for introducing both **R** and **G**? Why not train **G** directly from the null-space projection?
4. In Equation (3), why is **σ** omitted from the arguments of **R**? Is this intentional or an error?
5. Please clarify if the method is supposed to be a PnP method, and if not why the baseline is a PnP method which doesn't have access to the measurement operator at training time.
6. In the theoretical section, please improve the writing of your assumptions and theorems to make them understandable.
7. What exactly is the **Base** baseline in experiments? How was it implemented or chosen?

---

### Official Review · Reviewer_F891 · 2025-11-01

**Soundness:** 2
**Presentation:** 2
**Contribution:** 2
**Rating:** 2
**Confidence:** 4

**Summary:**

In this paper, the authors propose a Non-Linear Null-space Prior (NLNP) for solving inverse problems in computational imaging. The approach aims to explicitly model the null-space of the forward operator $H$, which corresponds to signal components invisible to the acquisition process. The proposed method jointly learns (i) an image prior through noise-conditional denoising, and (ii) a low-dimensional representation of the null-space components. During inference, a prior network predicts a null-space code from the noisy image, and a measurement encoder predicts the same code from the measurements. Reconstruction is performed by penalizing the mismatch between these predicted codes. The authors provide a theoretical justification via a projected Tweedie identity, showing that the network estimates the projected score of the data distribution, and claim that the regularizer introduces orthogonal, state-dependent curvature in the null-space of the sensing matrix, improving conditioning. The framework is integrated into a plug-and-play setup and tested on compressed sensing and image restoration tasks.

**Strengths:**

1. The method seems general enough to be applicable to various imaging settings, including compressed sensing and inpainting.
2. The proposed algorithm (adding to different prior gradients in the algorithm) is, to the best of my knowledge, novel and interesting.

**Weaknesses:**

1. The theoretical analysis is too limited and in particular not put in perspective of other recent works. (see suggestions in the next box)
2. Experimental results are insufficient. In particular, the authors do not provide baselines for comparisons, nor results on standard datasets. In this context, it is difficult to assess the method fairly.

**Questions:**

**Major comments:**
1. The theoretical analysis and proposed bounds are presented in isolation, without sufficient comparison to recent related works. Similar ideas (especially those involving projected score estimation and new Tweedie-like formulas) have been explored by Kamilov et al. (e.g., arXiv:2411.18970 and arXiv:2310.01391). The authors should clearly delineate how their contribution differs from or extends these studies.
2. The experimental section is underdeveloped and inconsistent with the claims. The paper frequently mentions compressed sensing, yet no visual results on such setups are provided. The authors should demonstrate their approach on standard benchmarks such as CBSD68, Urban100, or Set12, and ideally include color image results. Without visual and quantitative results on standard datasets and extensive comparisons with other relevant baselines, it is impossible to assess whether the proposed prior indeed provides better conditioning or improved reconstruction quality.

**Minor comments**
1. The term “non-linear” in the title and throughout the paper is unclear and potentially misleading. One would expect it to refer to non-linear inverse problems, yet the setup seems to remain linear with a non-linear learned prior. This should be clarified early on.
2. Lines 41–43: the notations around $x^∗$ and the range/null-space decomposition are not rigorous. $x^*$ is fixed, and the subspace
$\text{Span}(Hu), u \in \mathbb{R}^n$ should be explicitly defined as the range space for random vectors $u\in\mathbb{R}^n$.
3. line 043 - The noise is also responsible for ill-posedness (think for instance of Gaussian deblurring with nearly invertible kernel - the noise is responsible for the ill-posedness).
4. The statement “our key insight is to model, learn, and use the geometry of the null-space” is overstated. The geometry of the null-space of a linear operator is trivial and well known: it is an affine subspace (hyperplane) orthogonal to the range of $H^\top$. What is novel is the parametrization of null-space components through a learned code, not the geometry itself.
5. The claim that “every inverse problem solver seeks to determine the null-space component of $x$ only from $y$” is incorrect: this determination depends on both $y$ and $H$.
6. The approximation $\mathbb{E}[R(x,\sigma)∣y] \approx \mathbb{E}[Sx∣y]$ seems conceptually inaccurate. Isn't $R(x, \sigma)$ expected to estimate the full image over the space, not restricted to the range of $S$?
7. Line 260: an equation term appears to be missing.
8. The reference for total variation (Yuan, 2016) is outdated; a more appropriate reference would be a Pock et al. paper or other classical convex optimization sources.
9. The authors could remove Figure 2, move proofs to appendix, organize the Fig. 5 better: doing so would yield significant gains in space and allow the authors to detail much more related works and add other experimental results.

**Remarks:**
1. The constraint in equation (5), although working as was illustrated in Ryu et al., is not rigorous in the sense of the algorithm, as the rescaling operation is not a proper projection. A parametrization of W using unitary operators would be more appropriate if the goal is to preserve orthogonality. A more rigorous (and potentially more efficient during training) strategy would be to parametrize the operators as unitary operators.

---

### Official Review · Reviewer_EGmu · 2025-11-02

**Soundness:** 3
**Presentation:** 3
**Contribution:** 3
**Rating:** 4
**Confidence:** 4

**Summary:**

This paper proposed a Non-Linear Null-space Prior (NLNP) for inverse problems. In particualr, it constructs the prior by jointly learning two components: (1) a denoising subnetwork $W$, conditioned on noise, which learns the local geometry of the image manifold; (2) a network $R$, which learns and predicts the nonlinear encoding (nullspace code) of an image on a low-dimensional null-space slice $row(S)$. During training, $R$ outputs estimates of $Sx$ from noisy inputs, while a separate measurement encoder $G$ predicts the same code from measurements $y$. During reconstruction, mismatches between predicted and prior codes are incorporated as a regularisation term into a PnP-FISTA-style solver. The performance of NLNP is verified on image compressed sensing and deblurring tasks.

**Strengths:**

1. Overall, the idea is interesting and intutive, i.e. noise-conditioned training can couple the local manifold knowledge of $R$ with that of the denoiser $W$, may yielding more robust null-space estimates.

2. The author also demonstrated that the trained $R$ satisfies the ‘projected Tweedie identity’, such that $R$ estimates the projected score of the data distribution along the null-space slice direction. This injects ‘data-relevant’ curvature (second-order effects) into the null-space direction, thereby improving the condition number of the high-dimensional optimisation problem in unobservable directions without conflicting with data consistency ($range(H^T)$). An analysis of lower/upper bounds for the condition number under Gauss–Newton updates is also mentioned.

3. Experiments demonstrate performance improvements and acceleration effects under standard synthesis settings for toy applications (CS and deblurring).

**Weaknesses:**

1. IMO, the NLNP algorithm is sensitive to the choice of $S$ and its dimension $p$. Whilst the authors acknowledge that selecting $p$ involves trade-offs, they provide neither automated selection nor theoretical guidance (discussing only empirical trade-offs). Should $S$ fail to align with the task-relevant nullspace patterns, the code learned by $R$ may prove useless or even detrimental. I would welcome further discussion of this question by the authors, along with potential calibration strategies.

2. Although experiments demonstrated both performance improvements and acceleration effects under standard synthesis settings (CS and deblurring), the reconstruction results are not promising. Moreover, due to its supervised training, a comprehensive comparison can be made between existing supervised nullspace solvers for inverse problems (e.g. DDN, Chen et al., ECCV'20) and standard PnP-FISTA and diffusion-based PnP solvers (e.g. DiffPIR, Zhu et al., CVPRW'23). Finally, it's better to present some results on real & more challenging inverse imaging problems (e.g. accelerated MRI).

3. Some typos. Line 236: Typo '..'

**Questions:**

Could you please explain how the NLNP identifies the signal model, and whether the NLNP reconstruction is a one-to-one mapping? Also, can signals be uniquely recovered from their observations?

Regarding Eqs.(4), the author argued that $R^*$ derives curvature injection based on this form. What are the prerequisites for this equation to hold under strict probabilistic and partial differential conditions? For instance, does it require $p_\sigma$ to be smooth and R's functional space sufficiently large to express conditional expectations? Are there any boundaries or approximation terms that have been underestimated? Additional proof and numerical verification should be discussed.

Line 215: Figure 2. What is 'range invariance'? I see the NLNP training aims to ensure that $R$ maintains row or range invariance or predictability across different $x$ values corresponding to $y$. How is this guaranteed in practice? Should the training data distribution differ from the test distribution, IMO severe bias may arise.

Line 227: What is the $D_\sigma$, a pretrained DM denoiser? Should a hyperparameter for noise-level be involved in $D_\sigma$?

Figure 5: What do the numbers under each figure represent? PSNR/SSIM? If so, why do the NLNP reconstructions appear blurred? Is the high-frequency still underfitting? Could you also report the PSNR/SSIM for the first column?

---

### Meta-Review · Area_Chair_5Hr5 · 2026-01-07

**Summary:**

The authors propose a non-linear null-space prior (NLNP) for solving inverse problems in computational imaging, and integrate it into a plug-and-play (PnP) framework to solve problems in compressed sensing and deblurring. Some theory (showing that the prior improves the PnP algorithm performance) and simulated experiments are provided in support of the framework.

The reviews were negative. Issues with the current manuscript included (lack of) clarity in the writing, the very weak experimental results, and several questions/concerns about the methodology itself. For all these reasons, the manuscript is not suitable for publication yet.

**Reviewer Concerns:**

N/A, since there was no rebuttal.

**Reviewer Scores:**

N/A, since there was no rebuttal.

---

### Decision · Program_Chairs · 2026-01-26

Reject